# The Prognostic Value of One-Year Changes in Biventricular Mechanics for Three-Year Survival in Patients with Precapillary Pulmonary Hypertension: A Cardiovascular Magnetic Resonance Feature Tracking Study

**DOI:** 10.3390/medicina60010141

**Published:** 2024-01-12

**Authors:** Lina Padervinskienė, Joana Ažukaitė, Deimantė Hoppenot, Aušra Krivickienė, Paulius Šimkus, Irena Nedzelskienė, Skaidrius Miliauskas, Eglė Ereminienė

**Affiliations:** 1Department of Radiology, Medical Academy, Lithuanian University of Health Sciences, LT-44307 Kaunas, Lithuania; 2Faculty of Medicine, Medical Academy, Lithuanian University of Health Sciences, LT-44307 Kaunas, Lithuania; 3Department of Pulmonology, Medical Academy, Lithuanian University of Health Sciences, LT-44307 Kaunas, Lithuania; 4Department of Cardiology, Medical Academy, Lithuanian University of Health Sciences, LT-44307 Kaunas, Lithuania; 5Department of Radiology, Hospital of Lithuanian University of Health Sciences Kauno Klinikos, LT-50161 Kaunas, Lithuania; 6Department of Dental and Oral Diseases, Medical Academy, Lithuanian University of Health Sciences, LT-44307 Kaunas, Lithuania; 7Laboratory of Clinical Cardiology, Institute of Cardiology, Lithuanian University of Health Sciences, LT-50162 Kaunas, Lithuania

**Keywords:** pulmonary hypertension, cardiac magnetic resonance, right ventricle, feature tracking

## Abstract

*Background and Objectives*: The management of patients with pulmonary hypertension (PH) poses a considerable challenge. While baseline cardiac magnetic resonance imaging (cMRI) indices are recognized for survival prognosis in PH, the prognostic value of one-year changes in biventricular mechanics, especially as assessed using feature tracking (FT) technology, remains underexplored. This study aims to assess the predictive value of one-year change in cMRI-derived biventricular function and mechanics parameters, along with N-terminal pro-brain natriuretic peptide (NT-proBNP) levels and six-minute walking test (6MWT) results for three-year mortality in precapillary PH patients. *Materials and Methods*: In this retrospective study, 36 patients diagnosed with precapillary pulmonary hypertension (mPAP 55.0 [46.3–70.5] mmHg, pulmonary capillary wedge pressure 10.0 [6.0–11.0] mmHg) were included. Baseline and one-year follow-up cMRI assessments, clinical data, and NT-proBNP levels were analyzed. FT technology was utilized to assess biventricular strain parameters. Patients were categorized into survival and non-survival groups based on three-year outcomes. Statistical analyses, including univariate logistic regression and Cox regression, were performed to identify predictive parameters. *Results*: The observed three-year survival rate was 83.3%. Baseline right ventricle (RV) ejection fraction (EF) was significantly higher in the survival group compared to non-survivors (41.0 [33.75–47.25]% vs. 28.0 [23.5–36.3]%, *p* = 0.044), and values of ≤32.5% were linked to a 20-fold increase in mortality risk. RV septum longitudinal strain (LS) and RV global LS exhibited significant improvement over a one-year period in the survival group compared to the non-survival group (−1.2 [−6.4–1.6]% vs. 4.9 [1.5–6.7]%, *p* = 0.038 and −3.1 [−9.1–2.6]% vs. 4.5 [−2.1–8.5]%, *p* = 0.048, respectively). Declines in RV septum LS by ≥2.95% and in RV GLS by ≥3.60% were associated with a 25-fold and 8-fold increase in mortality risk, respectively. *Conclusions*: The decrease in right ventricular septal and global longitudinal strain over a one-year period demonstrates a significant predictive value and an association with an increased three-year mortality risk in patients with precapillary PH.

## 1. Introduction

Pulmonary hypertension (PH) is a progressive cardiovascular disorder, confirmed by right heart catheterization (RHC), when mean pulmonary arterial pressure (mPAP) is elevated at >20 mmHg at rest, according to the updated 2022 European Society of Cardiology and European Respiratory Society (ESC/ERS) PH Guidelines [1]. The prolonged elevation of pressure in the pulmonary arteries leads to an increased workload on the right ventricle (RV), which eventually causes right-heart failure and significantly affects patients’ mortality [2,3].

Various studies have demonstrated the significance of cardiac magnetic resonance imaging (cMRI) indices in predicting survival prognosis for patients with PH [3,4,5,6]. A recent study reported that risk assessment using cMRI parameters is as informative as RHC data and non-invasive cMRI could serve as a valuable tool for the follow-up evaluation of patients with PH [7]. In addition, the updated 2022 ESC/ERS PH Guidelines included several specific cMRI parameters in the risk stratification table as prognostic indicators [1]. Moreover, myocardial strain parameters calculated through a novel feature tracking (FT) function offer an enhanced assessment of myocardial contractility and deformation, providing a more precise evaluation of cardiac mechanics [8,9]. There is existing evidence supporting the possibility of using myocardial strain to detect subclinical biventricular dysfunction [8,9,10,11] and its use in early prediction of adverse prognosis [12]. Despite novel treatment options and early risk stratification parameters, the overall mortality of PH patients remains high [13,14]; therefore, ongoing research in this area is crucial for advancing our understanding and improving the management of this challenging cardiovascular disorder.

This study aims to evaluate the predictive significance of one-year changes in cMRI-derived biventricular function and mechanics parameters, along with N-terminal pro-brain natriuretic peptide (NT-proBNP) and six-minute walking test (6MWT) for three-year mortality in patients with precapillary PH.

## 2. Materials and Methods

### 2.1. Patient Selection and Testing

This retrospective study was conducted at the Hospital of Lithuanian University of Health Sciences Kauno klinikos. The study protocol complies with the ethical guidelines of the 1975 Declaration of Helsinki and on the 5 June 2015 was approved by the Regional Biomedical Research Ethics Committee of the Lithuanian University of Health Sciences (ID No. BE-2-23).

Patients with confirmed precapillary PH, diagnosed by RHC from November 2012 to October 2020, were selected from the hospital’s database. The study inclusion criteria included patients aged 18 years or older who gave informed consent; cMRI was performed at baseline before the initiation of the specific PH treatment and after a one-year follow-up period. Patients were excluded from the study based on the following criteria: absence of a one-year follow-up cMRI study, previous pulmonary artery endarterectomy before the one-year follow-up cMRI study, underlying cardiomyopathy, low-quality cMRI, and the absence of available data on the patient’s current health in the hospital database. Finally, 36 patients were enrolled in the study for a three-year survival analysis. The primary endpoints were cardiovascular death, and heart/heart–lung complex/lung transplantation. At the time of cMRI studies, all patients received specific PH treatment.

During the initial cMRI evaluation, we collected and analyzed various clinical data for the selected patients. This included age, sex, body surface area (calculated from height and weight), New York Heart Association functional classification (NYHA), mean pulmonary artery pressure (mPAP) from right heart catheterization, NT-proBNP level, and the results of the six-minute walking test (6MWT) [15].

### 2.2. Volumetric and Functional Measurements

cMRI scans were performed using a 1.5T whole-body system (Siemens Aera, Siemens Medical Solutions; Erlangen, Germany). Images were analyzed by an experienced radiologist using conventional CMR software (syngo.via; Siemens Healthcare). Four-chamber (4Ch) and short-axis (SA) cine images were captured using a retrospectively cardiac-gated multi-slice steady-state free precession (SSFP) sequence. The endocardial and epicardial surfaces were manually traced from the stack of axial images in the SA plane, covering both ventricles from base to apex. The stroke volume (SV) and ejection fraction (EF) of both ventricles were derived from the end-diastolic volume (EDV) and end-systolic volume (ESV), with measurements normalized to body surface area. Indexed EDV, ESV, and SV (EDVI, ESVI, and SVI, respectively) were used in the analysis. For right ventricle mass calculations, all RV papillary muscles were included, but the interventricular septum was considered as a part of the left ventricle (LV) [16]. In our previous work, we reported high inter- and intra-observer reproducibility [17].

### 2.3. Feature Tracking Mechanical Analysis

The myocardial strain was calculated using the cMRI FT software package (Medis Suite QStrain 3.2.0; Medis Medical Imaging Systems bv, Leiden, The Netherlands). Two-, three-, and four-chamber and short-axis cine images were imported into the software for the assessment of longitudinal strain (LS) for both ventricles and circumferential strain for the LV. The FT analysis was performed semi-automatically, as the endocardial surface contour was traced in the systole and diastole of the cardiac cycle. This process was meticulously examined and manually adjusted as needed. LV global longitudinal strain (GLS) was acquired by assessing strain in two-chamber, three-chamber, and four-chamber long-axis views and calculating the average strain. The LV global circumferential strain (GCS) entailed averaging the strain curves from the basal, mid, and apical segments acquired from the short-axis views. RV GLS was calculated using the cardiac four-chamber long-axis view.

### 2.4. Data Analysis

Statistical analysis was performed using the SPSS 29.0 package (SPSS, Chicago, IL, USA). Descriptive statistics are presented as numbers and percentages or median with interquartile range, as appropriate. A chi-square (χ2) test was used for qualitative data comparison. For continuous variables, the non-parametric Mann–Whitney U test was used to compare the two groups. A Wilcoxon Signed-Ranks Test was used to evaluate the parameter changes in groups. Calculated data are reported as a median with an interquartile range. For determining the optimal cut-off values of parameters, we utilized the receiver operating characteristic (ROC) curve approach, employing the Youden Index. In survival analysis, we first performed univariable logistic regression analysis, and the identified variables were included in a binary logistic regression model. Cox regression analysis was used to determine the hazard ratio (HR), and the distinction between matched groups was expressed with a 95% confidence interval (95% CI). Kaplan–Meir curves were designed to illustrate survival according to chosen threshold values. Two-tailed probability values at *p* < 0.05 were considered statistically significant.

## 3. Results

Patients were categorized into groups based on the three-year survival outcomes (survival and non-survival). Over the three-year observation period, six patients died due to cardiopulmonary complications (observed three-year survival—83.3%).

### 3.1. Baseline Clinical Characteristics

The majority of patients (*n* = 26; 72.2%) belonged to group 1 of the 2022 ESC/ERS Guidelines classification of PH [1], including subgroups of idiopathic pulmonary arterial hypertension (PAH) (*n* = 12; 33.3%), pulmonary hypertension associated with connective tissue disease (CTD-PAH) (*n* = 7; 19.4%), Eisenmenger syndrome (*n* = 6; 16.7%), and portopulmonic PH (*n* = 1; 2.8%). The remaining 10 patients (27.8%) were classified under group 4, experiencing non-operable chronic thromboembolic PH (CTEPH). The distribution of subgroups was consistent between the survival and non-survival groups (*p* = 0.605). At the evaluation, all patients received standard precapillary PH therapy, aligned with then-current guidelines and locally available medications, including phosphodiesterase type 5 inhibitors, endothelin receptor antagonists, prostacyclin analogs, and soluble guanylate cyclase stimulators. Most patients were on monotherapy (*n* = 23; 65.5%), while the rest were on combination therapy (*n* = 13; 35.5%). Treatment options did not differ between the survival and non-survival groups (*p* = 0.420).

The age of the non-survival group was significantly higher than that of the survival group (*p* = 0.015). Sex and NYHA functional class distribution did not differ between groups. The median baseline distance recorded during the 6MWT was higher in the survival group (*p* = 0.012). Although the *p*-value did not reach statistical significance, there was a noticeable trend indicating higher baseline NT-proBNP serum levels in the non-survival group (*p* = 0.096). Mean pulmonary artery pressure (mPAP), measured by RHC at the time of PH diagnosis, did not differ between groups (Table 1).

### 3.2. Baseline cMRI Parameter Evaluation

The baseline comparison study of cMRI parameters revealed a significantly higher RV EF within the survival group (*p* = 0.044), and there was a trend toward a higher RV SVI in this group of patients (*p* = 0.078). However, baseline biventricular mechanical strain parameters showed no differences between the groups (Table 2).

### 3.3. One-Year Clinical and cMRI Parameter Changes within and between Groups

Three-year survival group patients showed a tendency to improve LV GLS (*p* = 0.067), while the LV GCS increased significantly (*p* = 0.002) over a one-year period. Similar patterns were observed in RV strain parameters; RV free wall LS and RV GLS increased significantly (*p* = 0.043 and *p* = 0.049, respectively) and RV septum LS showed near-significant improvement by adjusting the sample size to *n* = 24 (*p* = 0.054). Within the survival group, only the LV stroke volume index (SVI) showed a tendency to increase over the follow-up period among the functional parameters (*p* = 0.058). In contrast to the survival patients, RV septum LS in the non-survival group patients tended to decline over the same period (*p* = 0.075), and no other functional or deformation parameters showed significant changes (Table 3).

The one-year change in mechanical right ventricle parameters (septum LS and GLS) exhibited a significant improvement in the survival group compared to non-survival group patients (*p* = 0.038 and *p* = 0.048, respectively). The RV mass increased in the survival group (*p* = 0.015), but there was no difference in RV mass between the groups (*p* = 0.419). There were no significant differences between the groups in terms of changes in LV volumetric and mechanical parameters, despite significant LV strain improvement in the survival group (Table 3).

### 3.4. Survival Analysis

In the survival analysis, initially, ROC curve analysis was conducted to identify threshold values for parameters that showed significance in the primary analysis (Table 4 and Figure 1).

Univariate analysis showed a baseline RV EF ≤ 32.5%, a reduction in RV septum LS by ≥2.95%, and a decline in RV GLS by ≥3.60% over a one-year period (Δ) (where a positive change indicates a decline in myocardial strain), which were associated with an increased risk of death, showing high specificity and sensitivity (Figure 1, Table 4 and Table 5). Even after adjusting for age, a significant association persisted between all analyzed predictor variables and mortality (Table 5).

Subsequently, univariate Cox regression analysis demonstrated a significant predictive value and association of baseline RV EF at the RV septum LS Δ and RV GLS Δ variables with the endpoints (Table 6 and Figure 2).

Finally, a bivariate logistic regression model was developed to predict death in three years, considering the baseline RV EF and RV GLS Δ cMRI parameters (Table 7).

## 4. Discussion

In this study, we aimed to assess the one-year changes in cardiac magnetic resonance imaging (cMRI)-derived biventricular function and mechanics parameters, alongside the levels of N-terminal pro-brain natriuretic peptide (NT-proBNP) and distance in the six-minute walking test (6MWT), as well as their predictive value in estimating three-year mortality among precapillary PH patients. Our findings provide important insights into the prognostic value of cMRI parameters in this population.

The observed three-year mortality in our study was 83.3%, consistent with findings from other studies. Benza R. L. et al. analyzed 2635 patients and reported the three-year survival rate to be 85% [13], while some studies describe survival rates as low as 67% [18].

Our study confirms the importance of RV function in the PH population. We found that baseline RV EF was significantly lower in the three-year non-survival group when compared to the survival group. Subjects in our non-survival group were classified as being at high risk for mortality, in line with all cMRI parameter criteria for risk stratification in the 2022 ESC/ERS PH Guidelines [1]. The especially low parameters observed could be explained by possible late diagnosis and already advanced disease with right ventricular dysfunction at the initial evaluation. We found a threshold value of ≤32.5% at baseline was associated with an increased risk of death risk in our study patients. In addition, the threshold value of RV EF < 25.5% was found to be a significant mortality prognostic factor in our previous research [17,19]. In contrast, Zhou et al. observed a higher cut-off point of RV EF < 40.4% associated with an increased hazard ratio for the combined endpoint [9]. A recent meta-analysis by Alabed et al. highlighted the importance of RV EF, as the pooled HR from 22 studies showed that every 1% decrease in RV EF is linked to a 2.1% higher risk of death within 54 months. Additionally, we observed no significant changes in RV EF over time in our population. This might be attributed to the short duration of the follow-up or the possibility that our small sample size failed to capture subtle changes. Therefore, our study highlights the critical role of right ventricular function in the prognosis of PH patients and underscores the need for more sensitive screening options, such as FT technology, in this patient population.

Another significant aspect of our study was the examination of biventricular strain parameters calculated using the FT function. Even though we did not find baseline biventricular myocardial strain differences between the analyzed study groups, the one-year changes were notable. We observed an improvement in LV mechanics in the survival group over time when compared to the non-survival group. Right ventricle free wall strain showed significant improvement over a one-year period in the survival group, but the change was not statistically different in comparison to that observed in the non-survival group, possibly due to the small sample size. An important finding is the change in RV septum LS and GLS over a one-year period, as they improved in patients that survived, while decreasing in the non-survival group. Also, those parameters were significantly associated with three-year mortality, and the associations persisted even after adjusting for age, highlighting the robustness of these predictors.

To our knowledge, no studies have published data about one-year changes in strain parameters yet. However, baseline strain values have been identified by other authors as crucial in predicting mortality in PH patients. Zhou D et al. observed lower baseline RV GLS and RV septum and free wall LS parameters in the clinical worsening group when compared to the no-endpoint group [9]. Other publications also describe an increased mortality risk associated with RV GLS deterioration [5,8,12,20]. Our findings align with existing evidence and underscore the importance of considering ventricular mechanics in assessing the overall cardiac function in precapillary PH patients. Furthermore, we show the significance of one-year changes in strain parameters for detecting subclinical dysfunction and predicting adverse prognosis.

Despite advances in treatment options and risk stratification parameters, the persistently high mortality rate among PH patients emphasizes the ongoing need for research in this field. Early detection of changes in biventricular mechanics may enable timely intervention and personalized treatment strategies, potentially improving patient outcomes. Currently, if a patient is receiving treatment and is stable, routine cMRI testing is not recommended [1]. However, as cMRI becomes more available, it could be beneficial to establish a standardized cMRI protocol to routinely screen patients for subclinical changes in cardiac function.

### Study Limitations

While our study provides valuable insights, certain limitations should be acknowledged. The study’s reliance on a single-center retrospective design and relatively small sample size may introduce selection bias and limit the generalizability of the findings to broader populations or healthcare settings. This study included patients from both group 1 and group 4, as classified by the 2022 ESC/ERS Guidelines for PH [1]. Although these groups differ in nature, combining them was necessary to achieve a sufficiently large sample size. This approach was essential for ensuring the statistical robustness of our analysis, a goal unattainable by examining these rare conditions separately. Future prospective studies with larger cohorts and longer follow-up periods are needed to validate our results and explore additional prognostic markers. Additionally, the software used for our feature tracking measurements is not designed to assess more detailed right ventricular parameters, such as RV circumferential or radial strain. Therefore, it was not possible to conduct a more thorough analysis of right ventricular mechanics, which could hold additional insights into RV deformation. Finally, many patients were excluded from the study as one-year follow-up cMRI was not performed. It is crucial to establish a standardized protocol for baseline and follow-up cMRI studies, which could be used by specialized PH centers for further research.

## 5. Conclusions

The baseline right ventricular ejection fraction and the reduction in right ventricular septal and global longitudinal strain over a one-year period are associated with an increased three-year mortality risk in precapillary PH patients. These findings contribute to the evolving landscape of risk assessment in PH patients and emphasize the potential role of novel FT technology in detecting subclinical ventricular dysfunction as compared to conventional cMRI data.

## Figures and Tables

**Figure 1 medicina-60-00141-f001:**
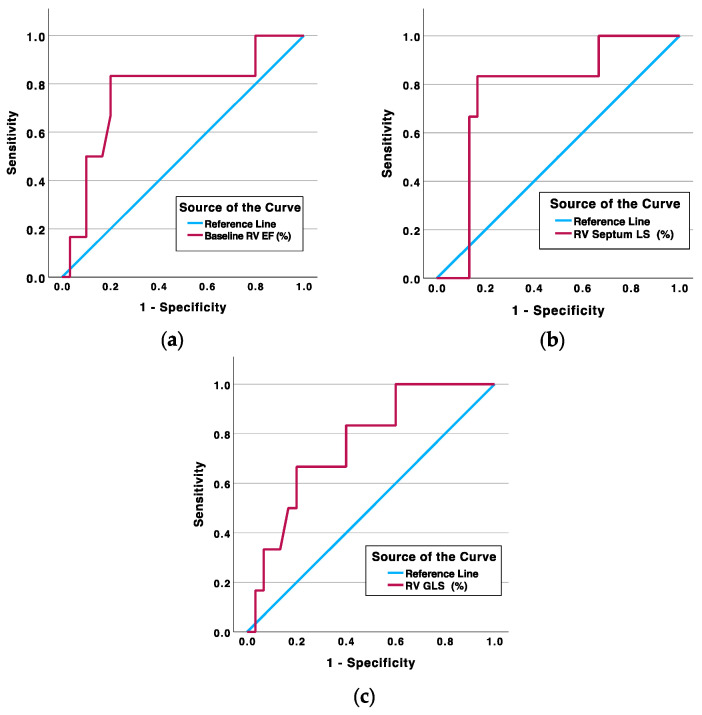
Receiver operating characteristic (ROC) curves of (**a**) baseline RV EF ≤ 32.5%, (**b**) RV septum LS Δ ≥ 2.95%, and (**c**) RV GLS Δ ≥ 3.60%. RV—right ventricle; GLS—global longitudinal strain, LS—longitudinal strain, EF—ejection fraction.

**Figure 2 medicina-60-00141-f002:**
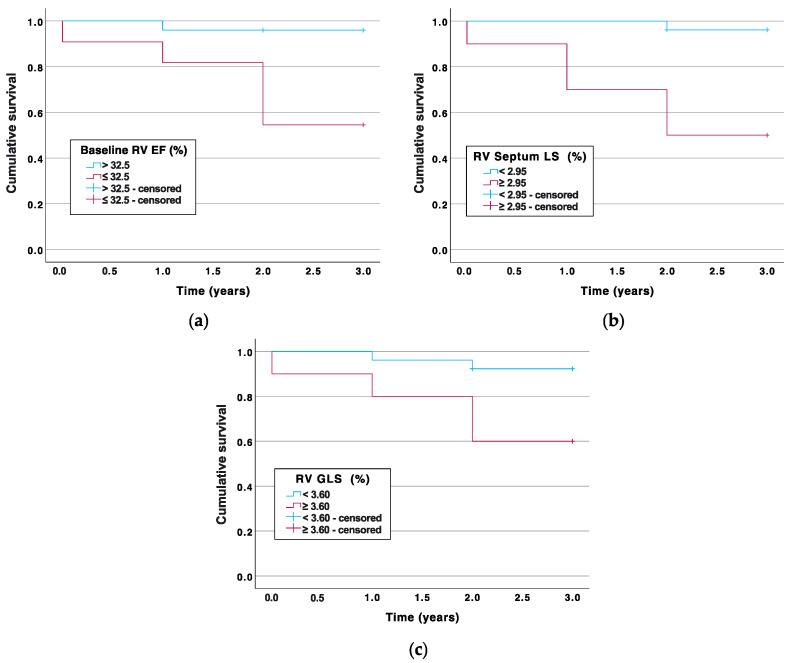
Kaplan–Meier curve estimation of the time to death according to (**a**) baseline RV EF ≤ 32.5%, (**b**) RV septum LS Δ ≥ 2.95%, (**c**) RV GLS Δ ≥ 3.60%. RV—right ventricle, GLS—global longitudinal strain, LS—longitudinal strain, EF—ejection fraction.

**Table 1 medicina-60-00141-t001:** Baseline clinical characteristics.

Parameter	All Patients (*n* = 36)	Survival Group (*n* = 30)	Non-Survival Group (*n* = 6)	*p*-Value
Age (years)	56.0 [45.3–68.8]	52.0 [41.8–63.3]	70.50 [56.5–72.3]	0.015 **
Women/men (N (%))	27/9 (75.0/25.0)	23/7 (76.7/23.3)	4/2 (66.7/33.3)	0.606 *
NYHA class 2/3/4 (N (%))	2/31/3 (5.6/86.1/8.3)	2/26/2 (6.7/86.7/6.7)	0/5/1 (83.3/16.7)	0.605 *
6MWT (m)	339.0 [210.0–414.0]	342.5 [261.8–420.0]	180.0 [116.0–240.0]	0.012 **
NT-proBNP (ng/L)	1472.0 [398.0–3484.0]	1433.0 [308.5–2915.5]	3131.5 [1308.0–6963.5]	0.096 **
mPAP (mmHg)	55.0 [46.3–70.5]	54.5 [45.8–67.5]	59.5 [49.3–80.3)	0.442 **

The *p*-values were determined using the chi-square (χ2) * test or Mann–Whitney Test **. Values are reported as N (%) or medians [interquartile range]. Definition of abbreviations: NYHA—New York Heart Association functional classification, 6MWT—six-minute walking test, NT-proBNP—brain natriuretic peptide, mPAP—mean pulmonary arterial pressure.

**Table 2 medicina-60-00141-t002:** Biventricular function and deformation parameters at baseline: cMRI measurements.

Parameter at Baseline	Survival Group (*n* = 30)	Non-Survival Group (*n* = 6)	*p*-Value between the Groups
LV GLS (%)	−17.9 [−24.2–(−15.4)]	−17.1 [−20.4–(−11.9)]	0.497
LV GCS (%)	−31.6 [−37.3–(−26.4)]	−31.1 [−38.3–(−23.8)]	0.799
LV EDVI (mL/m^2^)	64.0 [52.5–81.3]	58.0 [54.0–67.3]	0.384
LV ESVI (mL/m^2^)	28.0 [18.8–40.0]	26.5 [19.0–34.0]	0.702
LV SVI (mL/m^2^)	34.5 [31.0–45.0]	34.0 [29.5–38.3]	0.782
LV EF (%)	59.0 [48.5–65.5]	56.5 [41.5–66.5]	0.865
RV free wall LS (%)	−19.6 [−24.6–(−13.4)]	−15.0 [−19.5–(−12.7)]	0.243
RV septum LS (%)	−11.3 [−13.9–(−6.7)]	−12.4 [−16.3–(−7.5)]	0.552
RV GLS (%)	−14.0 [−16.6–(−10.8)]	−14.0 [−17.0–(−9.3)]	0.734
RV EDVI (mL/m^2^)	86.5 [64.8–108.8]	86.0 [69.5–111.5]	0.782
RV ESVI (mL/m^2^)	51.0 [33.3–69.8]	62.0 [45.0–88.5]	0.234
RV SVI (mL/m^2^)	33.0 [26.8–41.3]	25.0 [20.8–33.8]	0.078
RV Mass (g/m^2^)	46.5 [35.5–57.3]	50.0 [40.0–62.8]	0.432
RV EF (%)	41.0 [33.75–47.25]	28.0 [23.5–36.3]	0.044

The *p*-values were determined using the Mann–Whitney Test. Values are reported as medians [interquartile range]. Definition of abbreviations: LV—left ventricular, GLS—global longitudinal strain, GCS—global circumferential strain, EDVI—end-diastolic volume, ESVI—end-systolic volume index, SVI—stroke volume index, EF—ejection fraction, RV—right ventricular, LS—longitudinal strain.

**Table 3 medicina-60-00141-t003:** Changes in clinical and cMRI biventricular function and deformation parameters over a one-year period within and between study groups.

Parameter	Survival Group (*n* = 30/*n* = 24 *)		Non-Survival Group (*n* = 6)		*p*-Value between the Groups
	Δ	*p*-Value within the Group	Δ	*p*-Value within the Group	
6MWT (m)	15.0 [−25.0–52.5]	0.271	−60.0 [−100.0–] **	0.109	0.081
NT-proBNP (ng/L)	18.7 [−582.3–362.0]	0.495	481.0 [−183.5–4576.0]	0.138	0.098
LV GLS (%) ***	−1.35 [−8.9–2.1]	0.067	−1.8 [−21.1–5.3]	0.463	0.832
LV GCS (%) ***	−2.7 [−8.4–(−0.5)]	0.002	−2.8 [−3.7–4.4]	0.463	0.396
LV EDVI (mL/m^2^)	1.5 [−1.0–6.5]	0.101	0.0 [−11.3–19.0]	0.674	0.882
LV ESVI (mL/m^2^)	0.0 [−4.0–4.0]	0.810	2.5 [−6.0–18.0]	0.600	0.457
LV SVI (mL/m^2^)	4.0 [−1.5–7.3]	0.058	−1.5 [−20.3–7.3]	0.600	0.327
LV EF (%)	2.0 [−6.8–8.3]	0.532	−1.0 [−22.8–8.5]	0.600	0.339
RV free wall LS (%) ***	−3.5 [−9.3–3.8]	0.043	−0.45 [−6.6–5.9]	0.917	0.445
RV septum LS (%) ***	−1.2 [−6.4–1.6]/ −1.8 [−6.1–0.8]	0.165/ 0.054	4.9 [1.5–6.7]/ 4.9 [1.5–6.7]	0.075/ 0.075	0.038 0.020
RV GLS (%) ***	−3.1 [−9.1–2.6]/ −3.6 [−8.7–1.6]	0.049/ 0.027	4.5 [−2.1–8.5]/ 4.5 [−2.1–8.5]	0.249/ 0.249	0.048 0.028
RV EDVI (mL/m^2^)	0.5 [−7.0–10.8]	0.737	8.0 [−7.3–17.8]	0.400	0.470
RV ESVI (mL/m^2^)	3.0 [−5.0–11.0]	0.179	11.5 [−3.8–19.3]	0.173	0.298
RV SVI (mL/m^2^)	−2.0 [−7.0–5.8]	0.436	−0.50 [−8.0–10.5]	0.752	0.766
RV Mass (g/m^2^)	3.5 [−2.3–9.3]	0.015	8.0 [−1.5–15.0]	0.136	0.419
RV EF (%)	−1.5 [−8.0–5.0]	0.249	−2.5 [−9.8–8.0]	0.753	0.949

* To improve statistical accuracy, a subset of 24 randomly selected individuals were used. ** Third quartile is not reported due to small sample size (*n* = 3). *** Greater negative values indicate improvement. The *p*-values were determined using the Mann–Whitney Test and Wilcoxon Signed-Ranks Test. Values are reported as medians [interquartile range]. Definition of abbreviations: Δ—one-year change in the parameter, 6MWT—six-minute walking test, NT-proBNP—brain natriuretic peptide, LV—left ventricular, GLS—global longitudinal strain, GCS—global circumferential strain, EDVI—end-diastolic volume, ESVI—end-systolic volume index, SVI—stroke volume index, EF—ejection fraction, RV—right ventricular, LS—longitudinal strain.

**Table 4 medicina-60-00141-t004:** Risk of death based on univariable analysis using ROC test.

Parameter (Threshold Value)	Area Under the Curve (%)	Sensitivity/Specificity (%)	Survival/Non-Survival (*n* (%))	*p*-Value
Baseline RV EF ≤ 32.5%	76.4	83.3/80.0	6/5	0.006
RV septum LS Δ ≥ 2.95%	77.2	83.3/83.3	5/5	0.003
RV GLS Δ ≥ 3.60%	76.1	66.7/80.0	6/4	0.039

Δ—one-year change in the parameter, RV—right ventricle, GLS—global longitudinal strain, LS—longitudinal strain, EF—ejection fraction.

**Table 5 medicina-60-00141-t005:** The risk of death based on univariate binary logistic regression analysis.

Parameter (Threshold Value)	Odds Ratio [95% CI]	*p*-Value	Odds Ratio Adjusted by Age [95% CI]	*p*-Value
Baseline RV EF ≤ 32.5%	20.0 [1.954–204.728]	0.012	22.067 [1.739–279.963]	0.017
RV septum LS Δ ≥ 2.95%	25.0 [2.380–262.653]	0.007	27.034 [1.982–368.747]	0.013
RV GLS Δ ≥ 3.60%	8.0 [1.174–54.497]	0.034	13.989 [1.243–157.400]	0.033

Δ—one-year change in the parameter, RV—right ventricle, GLS—global longitudinal strain, LS—longitudinal strain, EF—ejection fraction.

**Table 6 medicina-60-00141-t006:** Risk of death based on univariate Cox regression analysis.

Parameter (Threshold Value)	Hazard Ratio [95% CI]	*p*-Value	Hazard Ratio Adjusted by Age [95% CI]	*p*-Value
Baseline RV EF ≤ 32.5%	12.721 [1.485–108.932]	0.020	9.883 [1.109–88.072]	0.040
RV septum LS Δ ≥ 2.95%	16.513 [1.924–141.767]	0.011	14.744 [1.686–128.925]	0.015
RV GLS Δ ≥ 3.60%	5.996 [1.097–32.775]	0.039	7.210 [1.303–39.895]	0.024

Δ—one-year change in the parameter, RV—right ventricle, GLS—global longitudinal strain, LS—longitudinal strain, EF—ejection fraction.

**Table 7 medicina-60-00141-t007:** Bivariate logistic regression analysis model.

Regressors	OR [95% CI], *p*-Value
Model No. 1 (correct prediction 88.9%, Nagelkerke determination coefficient 0.527)	
Baseline RV EF ≤ 32.5%	28.163 [1.836–431.973], 0.017
RV GLS Δ ≥ 3.60% *	12.285 [0.96–156.905], 0.054
Model constant	−4.419, *p* = 0.0.12

* A greater value indicates a decline in myocardial strain. Δ—one-year change in the parameter, RV—right ventricle, GLS—global longitudinal strain, EF—ejection fraction.

## Data Availability

The datasets used and analyzed in this study are available from the corresponding author upon reasonable request.

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
