# Peer review of "The Prognostic Value of One-Year Changes in Biventricular Mechanics for Three-Year Survival in Patients with Precapillary Pulmonary Hypertension: A Cardiovascular Magnetic Resonance Feature Tracking Study"

_medicina, 2024, doi:10.3390/medicina60010141_

Round 1

Reviewer 1 Report

Comments and Suggestions for Authors

The study is of really great value, proving once again the need for new prognostic markers in PAH. It seems that regular patients' assessment parameters such as 6MWT are not sufficient.

I am not in favor of combining patients with CTEPH and PAH, these diseases differ in pathophysiology and probably a different response of the right ventricle to increasing pulmonary resistance, but I understand the need of higher number of patients. Please add this issue to the discussion. 

The authors focused mainly on RV GLS, it is worth citing the following work on this topic: 

Kazimierczyk, R., Malek, L.A., Szumowski, P. et al. Multimodal assessment of right ventricle overload-metabolic and clinical consequences in pulmonary arterial hypertension. J Cardiovasc Magn Reson 23, 49 (2021). https://doi.org/10.1186/s12968-021-00743-2

The manuscript is of a scientific nature, very well written stylistically and linguistically. The statistical methods were well selected. I really appreciate the aptly described study limitations.

Congratulations!

Author Response

Dear reviewer,

Thank you for your valuable insights and comments. We appreciate the opportunity to clarify and enhance our manuscript in response to the points you've raised. Please find the detailed responses below and the corresponding corrections highlighted in the re-submitted files.

Comment 1: I am not in favor of combining patients with CTEPH and PAH, these diseases differ in pathophysiology and probably a different response of the right ventricle to increasing pulmonary resistance, but I understand the need of higher number of patients. Please add this issue to the discussion.

Response 1: Thank you for pointing this out. We agree with your opinion, as these two conditions, despite being in the precapillary PH group, can differ. We have added the comments in the limitations sections. Please see the addition in lines 324-328 of the manuscript. 

Comment 2: The authors focused mainly on RV GLS, it is worth citing the following work on this topic: Kazimierczyk, R., Malek, L.A., Szumowski, P. et al. Multimodal assessment of right ventricle overload-metabolic and clinical consequences in pulmonary arterial hypertension. J Cardiovasc Magn Reson 23, 49 (2021). https://doi.org/10.1186/s12968-021-00743-2

Response 2: Thank you for the suggestion. We have analyzed the following article and added its citation.

Reviewer 2 Report

Comments and Suggestions for Authors

The article is relevant and has an important clinical value.

Myocardial strain identifies global and regional abnormalities in myocardial function. The most widely available and commonly used technique is echocardiography.

The gold standard is cardiac magnetic resonance imaging (CMRI). It is highly accurate and reproducible and can assess both structure and function. But strain can also be measured on CMRI.

Usually myocardial strain can be longitudinal, circumferential, and radial.

However, there are small comments.

1. Low statistical power of the study, in one of the groups of only 6 patients.

2. It is necessary to indicate what therapy the patients received.

3. How can you explain the initially low EF of the right ventricle?

4. And the fact that EF did not significantly change in one year?

Author Response

Dear reviewer,

Thank you very much for the time and effort you have invested in reviewing our manuscript. We are grateful for kind words and insightful suggestions on how to improve the manuscript. Please find the detailed responses below and the corresponding corrections highlighted in the re-submitted files.

Comment 1: Low statistical power of the study, in one of the groups of only 6 patients.

Response 1: Thank you for the comment. We have already acknowledged the small sample size in the limitation section, as this study includes patients with a rare disease and one-year dynamics evaluation makes it challenging to have a larger sample size. To improve statistical accuracy, we have performed additional analysis of significant parameters identified in the full-sample analysis. We randomly selected a subset of 24 individuals from the survival group and results remained statistically significant (please see Table 3). 

Comment 2: It is necessary to indicate what therapy the patients received.

Response 2: We have added specific groups of medications, which are used by our patients in monotherapy or combination therapy regiments. Please see the addition in lines 154-158 of the manuscript.

Comment 3: How can you explain the initially low EF of the right ventricle?

Response 3: In our opinion, the particularly low RV EF could be attributed to the typically late diagnosis, experienced by many PH patients. Consequently, the disease is already advanced, leading to right ventricular dysfunction at the initial evaluation. Furthermore, the median age of the non-survivor group was higher compared to the survivor group, potentially contributing to a worse clinical presentation. Nonetheless, in the survival analysis, strong associations persisted between higher mortality and lower baseline RV EF after adjusting for age. This confirms the significant impact of RV EF alone on survival. We have included these explanations in the discussion section, specifically in lines 272-276 of the manuscript. Thank you for highlighting these points.

Comment 4: And the fact that EF did not significantly change in one year?

Response 4: Thank you for this observation. We assume that a one-year period might be too short to observe a significant change in RV EF. In contrast, more sensitive strain parameters do show significant dynamics within this timeframe. Additionally, the small sample size might contribute to the lack of change detected in this particular study. However, there still appears to be a trend towards a decline in RV EF, likely due to the chronic nature of this condition, which limits the potential for improvement. Please refer to the addition in lines 283-288 of the manuscript.